# Effect of Brown Seaweed (*Macrocystis pyrifera*) Addition on Nutritional and Quality Characteristics of Yellow, Blue, and Red Maize Tortillas

**DOI:** 10.3390/foods11172627

**Published:** 2022-08-30

**Authors:** Alexa Pérez-Alva, Diana K. Baigts-Allende, Melissa A. Ramírez-Rodrigues, Milena M. Ramírez-Rodrigues

**Affiliations:** 1Department of Chemical, Food and Environmental Engineering, Universidad de las Américas Puebla, Ex Hacienda Sta. Catarina Mártir, Cholula 72810, Puebla, Mexico; 2Faculty of Agrobiology, Czech University of Life Sciences Prague, Food and Natural Resources, Kamýcká 129, 16500 Prague, Czech Republic; 3Food Analysis Laboratory, Intema S.A. de C.V., 31 Sur 2901, Col. Santa Cruz Los Ángeles, Puebla 72400, Puebla, Mexico; 4Tecnologico de Monterrey, Department of Bioengineering, Vía Atlixcáyotl 5718, Reserva Territorial Atlixcáyotl, Puebla 72453, Puebla, Mexico

**Keywords:** brown seaweed, *Macrocystis pyrifera*, maize, *tortillas*, phenolic compounds, antioxidant capacity

## Abstract

The objective of this study was to analyze the effect of incorporating *Macrocystis pyrifera* into yellow, blue, and red maize *masa* and *tortillas*. The nutritional composition and mineral content of *tortillas* was determined, and the color, texture, total phenolic compounds (TPC), and antioxidant capacity of *masas* and *tortillas* were measured. The addition of seaweed led to a significant decrease in moisture and a significant increase in ash, protein, and fiber, while no differences were observed in the lipid and carbohydrate content. There was a significant increase in all analyzed minerals (Na, Ca, P, K, and Mg). *Tortillas* weighed 24.54 ± 1.02 g, had a diameter of 11.00 ± 0.79 cm, and a thickness of 0.32 ± 0.09 cm. All color parameters were significantly affected by seaweed concentration. The hardness of the *masas* was 2.18–22.32 N, and the values of the perforation test of the *tortillas* were 1.40–4.55 N. The TPC of the *masas* and *tortillas* was measured in water and methanol:water extracts. Results were higher in the water extracts (1141.59–23,323.48 mg GAE/100 g *masa* and 838.06–2142.34 mg GAE/100 g *tortilla*). Antioxidant capacity (ORAC) was higher for methanol:water extracts (14,051.96–44,928.75 µmol TE/100 g *masa* and 14,631.47–47,327.69 µmol TE/100 g *tortilla*).

## 1. Introduction

Cereals and legumes are primary staple foods and key components of the human diet [1,2]. Specifically, cereals are the most important food source, and cereal-based foods provide a large part of the global population’s energy, protein, B vitamins, and minerals [1]. Among them, maize or corn (*Zea mays* L.), an annual crop from the family *Poaceae,* had the highest worldwide production in 2019 with 1,148,487,291 tons [3,4]. During the same year, the world production of wheat and rice were 765,769,635 and 755,473,800 tons, respectively [3].

In Mexico, maize is considered the main staple food [5], with a production of 27,228,242 tons in 2019 [6], which places it as the eighth-largest producer of maize in the world [7]. Additionally, Mexico possesses the largest diversity of genetic resources of maize in the world, with about 59 different landraces. There are many different types of maize, such as flour maize, flint maize, dent maize, sweet maize, popcorn, waxy maize, and amylomaize [8].

Several types of pigmented genotypes exist. Colors vary from white or yellow to violet, red, black, and blue [9], with blue and red being the most common pigmented genotypes [10]. Tortillas are the most consumed maize-based product in Mexico, and have been the basis of the daily diet for centuries [5,11]. The origin of maize *tortillas* dates back to Mexico’s prehispanic society, and, in some rural areas of the country, their preparation method remains the same even nowadays [12]. Other countries in which maize *tortillas* are consumed are Guatemala and the United States [13].

The nutritional value of maize can be affected by the variety, environment, and sowing conditions [14]. Maize is often subjected to nixtamalization, which consists of cooking the grains in an alkaline solution [5]. This process increases the bioavailability of calcium and niacin (B3), reducing the presence of mycotoxins [15,16]. Generally, it is considered a source of many vitamins, especially vitamins C, E, K, and B complex (B1, B2, B3, B5, B6, B9), as well as ß-carotene, selenium, and potassium [2,4]. However, its protein content is considered to be of low nutritional quality due to the low concentration of lysine and tryptophan [14]. 

*Tortillas* are an excellent vehicle to enhance nutrition options for maize consumers [17]. However, some studies have reported nutritional differences due to both pigmented genotypes [10] and the incorporation of different ingredients such as grasshopper (*Sphenarium purpuracens)* [18], nopal [11], faba bean (*Vicia faba*), white bean (*Phaseolus vulgaris*) [14], and *muicle* (*Justicia spicigera* Schechtendal) [5], among others. As a result, modifications to the mineral [11,18], protein [14,18], and dietary fiber [11,14] content, as well as the antioxidant capacity [5], have been observed. 

Over the last decades, seaweeds have gained attention as potential sources for the elaboration of food and feed products [19,20]. Seaweeds are marine, photosynthetic algae that are prolific in every ocean. They are divided into three main classes, or phyla: Phaeophyta (brown algae), Rhodophyta (red algae), and Chlorophyta (green algae) [21]. Depending on the phyla, the chemical composition of the seaweeds may vary. In general, they can present a protein content between 5 and 47% (dry mass), while the carbohydrates can constitute from 4 to 76% (dry mass), and the lipid content 0.60–4.14% (dry mass). Additionally, they are an important source of vitamins and minerals, with some species having up to 100 times more minerals and vitamins per unit dry mass than terrestrial plants or animal-derived foods [21]. Brown algae are rich in unsaturated fatty acids, biopolymers such as alginate, and phlorotannins (polymeric structures of phloroglucinol) [22]. *Macrocystis pyrifera* is a brown seaweed found in the Pacific Ocean, and is mainly used as feed for abalone and hydrocolloid production [19]. It has been reported that *M. pyrifera* contains ~60% carbohydrates, ~10% protein, and ~1.50% lipid content, with the remaining ~30% corresponding to ashes [19]. This seaweed has been used to prepare fritters and breadsticks, with the results showing an improvement in the fatty acid content [20]. 

Therefore, this study analyzed the effect of incorporating the brown seaweed *Macrocystis pyrifera* into maize *tortillas*. The seaweed was used at four different concentrations (including a control), and three different maize genotypes (yellow, blue, and red) were tested. As part of the analysis, the nutritional composition, color, texture, mineral content, phenolic compounds, and antioxidant capacity were determined. 

## 2. Materials and Methods

### 2.1. Test Materials, Chemicals, and Standards

Three different varieties (yellow, blue, and red) of nixtamalized *masa* (*Zea mays*) were purchased in a local store in Puebla, Mexico. The seaweed *Macrocystis pyrifera* was bought from ALGAS PACIFIC (Ensenada, Baja California, Mexico). The reagents used for the nutritional composition, total phenolic content, and antioxidant capacity were of analytic grade and were purchased from Sigma-Aldrich Co. (St. Louis, MO, USA). 

### 2.2. Tortilla Elaboration

To prepare the *tortillas*, the seaweed was first dried at 60 °C for 4 h, then milled and sieved with a 270 µm mesh. The seaweed powder was then incorporated into the nixtamalized *masas.* The seaweed was added at four different concentrations (0, 3, 6, and 9% (*w*/*w*)) into the three different varieties of *masa* (yellow, blue, and red) (Table 1). The seaweed concentration was determined based on preliminary experiments. To fully incorporate the seaweed powder, the mix was thoroughly kneaded. After obtaining an homogeneous dough (called *masa*), each *masa* was divided into 30 g spherical portions, which were then pressed into flat discs and heated in a hot griddle at 350 °C for 1 min, turned over and heated for 30 s on the other side, and heated again for 15 s on the initial side, as reported by Alvarez-Poblano et al. [5]. The same codes used for *masa* samples were applied for *tortillas* type identification by changing the M to T. The texture was analyzed in warm and room temperature (RT, 25 °C) *tortillas.* At the same time, color was determined only at RT. After these determinations, the remaining *tortillas* were milled in order to obtain a homogeneous product, which was frozen (−20 °C) until further analysis.

### 2.3. Nutritional Composition

The chemical composition of *tortillas* was determined according to the official AOAC methods for protein (954.01), lipids (920.39), total dietary fiber (962.09), ash (923.03), and moisture (925.09) content [23]; carbohydrates were calculated by difference. The protein conversion factor used to determine the protein content was 6.25.

### 2.4. Mineral Profile

The mineral profile (Na, Ca, P, K, and Mg) of *tortillas* was determined by Inductively Coupled Plasma-Optical Emission Spectrometry (ICP-OES) based on AOAC Method 2011.4, as reported by Kumaravel & Alagusundaram [24]. First, 0.5 g of the sample was digested with 10 mL of nitric acid (HNO_3_) using microwave digestion on a CEM Mars6 microwave system (Charlotte, NC, USA). The mixture was placed into an inert polymeric microwave vessel, which was sealed and heated for 60 min. After the digestion, the solutions were cooled and diluted to 50 mL with triple distilled water. The determination of mineral contents in this clear solution was carried out in triplicate using an Agilent 5110 ICP-OES (Santa Clara, Ca, USA). The analytical measurements were made with an ICP Expert 7.5.1 software equipped with a peristaltic pump, a crossflow nebulizer (coupled to a ryton double pass spray chamber), and a ceramic central torch tube injector with an internal diameter of 2 mm. The wavelengths used were 589.592, 315.886, 214.914, 766.491, and 280.27 for Na, Ca, P, K, and Mg, respectively. A five-point calibration curve prepared using a multielement standard solution (10–50 mg/L Merck KGaA, Darmstadt, Germany) was used.

### 2.5. Quality Characteristics

The surface color of randomly selected spots of *masa* and *tortilla* was measured in triplicate using a Konica Minolta CR-400 colorimeter (Konica Minolta Holdings Inc., Tokyo, Japan). To ensure the uniformity of the measuring conditions, the measurements were done in a portable photo booth (PULUZ Technology Ltd., Shenzhen, China) under white light and on a white background. The colorimeter was positioned with a 0° viewing angle geometry. The color was expressed as L*, a*, and b* colorimetric coordinates in the CIELab scale, as reported by Méndez-Lagunas et al. [25]. The hue angle (h), chroma (C*), and ΔE were calculated according to Equations (1)–(3) respectively [26].
(1)$$H=tan^{-1}\left(\frac{b^{*}}{a^{*}}\right)$$
(2)$$C^{*}=\sqrt{{\left(a^{*}\right)}^{2}+{\left(b^{*}\right)}^{2}}$$
(3)$$\Delta E=\sqrt{\left(\Delta L^{*2}+{\Delta a}^{*2}+{\Delta b}^{*2}\right)}$$

As part of the *tortillas’* characterization, their diameter and thickness were analyzed. The quantification consisted of an average of two diagonal measurements of three different *tortillas*, while thickness was determined by measuring five stacked *tortillas* and dividing the result by five [27]. To assess their rollability, warm and RT *tortillas* were cut into 2 cm wide strips. Next, each strip was wrapped around a wooden cylinder that was 2 cm in diameter, and the degree of rupture was observed. Rollability was rated on a scale of 1 to 5, where 1 = no breakage, 2, 3, 4, and 5 = 25%, 50%, 75%, and 100% breakage, respectively [28,29]. The texture of RT *tortillas* was measured as a perforation test, which consists of applying the necessary force to cause the *tortilla* to break, as reported by Argüello-García et al. [28]. The hardness of the *masas*, described as maximum force of the first compression (N) [18], was done following the procedure described by Montemayor-Mora et al. [27]. The texture analyses of *masas* and *tortillas* were done using a texturometer Shimadzu EZ-SX Texture Analyzer (Shimadzu Co., Kyoto, Japan) and the software used for data analysis was Trapezium X (Shimadzu Co.).

### 2.6. Total Phenolic Content and Antioxidant Capacity

The extracts used to quantify total phenolic content (TPC) and antioxidant capacity were done by diluting 1 g of either *masa* or ground *tortilla* in 10 mL of distilled water or in 10 mL of the methanol:water (90:10 (*v*/*v*)) solution. After 12 h, samples were centrifuged at 1325× *g* for 10 min. TPC was measured using a UV-Vis spectrophotometer model Multiskan Sky Microplate (Thermo Fischer Scientific, Waltham, MA, USA). TPC was measured using the Folin–Ciocalteu assay according to the method described by Xiang et al. [30]. First, 1 mL of distilled water was mixed with 0.20 mL of the sample, standard, or water (blank) and 0.25 mL of Folin–Ciocalteu reagent. After six minutes, 2 mL of water, and 2.5 mL of a 7% Na_2_CO_3_ solution were added. The tubes stood in the dark for 90 min, and their absorbance was read spectrophotometrically at 760 nm. TPC was calculated with a gallic acid calibration curve (0–1000 mg/L) and expressed as gallic acid equivalents (mg GAE/100 g).

The antioxidant capacity was measured as Oxygen Radical Absorbance Capacity (ORAC) using a Synergy HTX Multimode Reader (Bio Tek, Winooski, VT, USA). ORAC was determined following the method described by Huang et al. [31]. A stock solution of fluorescein (4 µM) was prepared in a 75 mM phosphate buffer (pH 7.4) and stored wrapped in foil at 4 °C. Before the assay, the stock solution was diluted (1:500 *v*/*v*) using 75 mM phosphate buffer (pH 7.4); the 75 mM phosphate buffer (pH 7.4) was also used to dissolve 2,2′-Azobis(2-methylpropionamidine) dihydrochloride (AAPH). The assay was done using a 96-well plate. In each well, 150 µL of the sodium fluorescein working solution was added, followed by 25 µL of phosphate buffer (blank), Trolox (curve), or sample. After incubation at 37 °C for 30 min, 25 µL of AAPH solution was added to each well. The fluorescence was then monitored kinetically with data taken every minute for 6 h and results were expressed as Trolox equivalents (µmol TE/100 g).

### 2.7. Statistical Analysis

Each measurement was repeated in triplicate and results were expressed as mean ± SD. The statistical analyses were carried out using Minitab 19 Statistical software (Minitab, LLC, State College, PA, USA) and consisted of a two-way analysis of variance (ANOVA) to identify the effect of the seaweed addition, the type of maize, and their interaction; a one-way ANOVA and a Tukey test were used to determine the differences among samples. All tests were performed using α = 0.05. All results showed a CV lower than 5%.

## 3. Results and Discussion

### 3.1. Nutritional Composition of Tortillas 

Significant differences between both colored maize and different concentrations of seaweed were observed (Table 2). For moisture, ash, protein, lipids, and fiber, both the type of maize and the concentration of seaweed used showed a significant effect (*p* < 0.05). However, the interaction between the two factors was not significant (*p* > 0.05). The concentration of seaweed used and the interaction between the latter and the type of maize were not significative (*p* > 0.05) for the carbohydrate content, and only the type of maize was significative (*p* < 0.05). The moisture content of the *tortillas* ranged between 40 and 46%. It can be observed that all control *tortillas* showed higher moisture values than *tortillas* with different concentrations of seaweed but the same type of maize, with the *tortilla* with blue maize (BT0) showing the highest moisture value (46.7%). It can also be noted that the moisture value decreased with an increase in seaweed. In general, *tortillas* with blue maize (BT0-BT9) showed the highest values, while the lowest values were reported for *tortillas* with yellow maize (YT0-YT9). It has been reported that the moisture of *tortillas* is between 40 and 50% [32]. While Alvarez-Poblano et al. [5] reported values ranging between 48 and 60% for *tortillas* with different concentrations of *muicle*, Argüello-García et al. [28] reported values between 44 and 47% for *tortillas* fortified with nontoxic *Jatropha curcas* flour. Both groups reported an increase in moisture when there was an increase in *muicle* or *J. curcas* flour, which is in opposition to the findings of this study. The ash content was between 1.12 and 4.42%, with BT9 showing the highest value (4.42%). The increase in seaweed concentration led to a significative increase in ash content. While there were no differences between yellow and red maize *tortillas*, all blue samples presented higher values. The ash content of dried *Macrocystis pyrifera* is generally reported to be between 22 and 36% [33], which could explain why an addition of 9% seaweed led to an increase of almost four times the mineral content of the control samples. Other authors have also reported an increase in the mineral content when *tortillas* are fortified [28]. 

There was also a significative increase in the protein content related to the increase in seaweed concentration. Values ranged from 4.49% (YT0) to 5.08% (RT9). *Tortillas* prepared with red maize presented the highest protein content, followed by blue and, subsequently, yellow maize. The lipid content was between 0.36 and 0.62%, with RT0 presenting the highest value. The incorporation of seaweed did not represent a significant difference among each type of maize. Moreover, as observed for the protein content, red maize presented higher lipid values, followed by blue maize, and, subsequently, yellow maize. The content of the protein and lipids in *tortillas* from this study was lower than the one reported for *tortillas* with different levels of fortification with *J. curcas* flour (6.52–10.85 and 3.35–3.95%, respectively) [28]. The carbohydrate content was not affected by the incorporation of seaweed (46.9–49.9%). However, *tortillas* prepared with yellow maize presented significantly higher values, followed by red, and, subsequently, blue maize. Despite the carbohydrate content not being modified, the incorporation of seaweed significantly increased the total fiber content. The values were between 6.72 and 11%, with RT9 being the sample with the highest result. Red maize *tortillas* presented higher values, followed by yellow maize *tortillas* and blue maize *tortillas*. The results obtained in this study were almost ten times higher than the ones for *tortillas* with different levels of fortification with *J. curcas* flour (0.8–1.98%) [28]. The incorporation of *Laminaria ochroleuca* in pasta also led to an increase in the insoluble fraction of dietary fiber, which shows that seaweeds can be used as texturing and bulking agents in different food products [34]. According to the Dietary Guidelines for Americans [35], the recommendation for fiber intake is between 20– 30 g per day, which could be obtained with three *tortillas*.

### 3.2. Mineral Profile of Tortillas

The mineral content of *tortillas* is shown in Table 3. The incorporation of seaweed led to significant differences in all analyzed minerals (Na, Ca, P, K, and Mg). The type of maize, the concentration of seaweed, and the interaction of both factors was significant (*p* < 0.05) in all analyzed minerals except for Ca, where the interaction between the factors was not significant (*p* > 0.05). The Na concentration ranged between 3.67 mg/100 g (RT0) and 303.07 mg/100 g (YT9), Ca was 100.14–612.26 mg/100 g (RT0 and BT9, respectively), the P content ranged between 78.39 mg/100 g (RT0) and 329.07 mg/100 g (YT9), the K content was 53.35–1700.62 mg/100 g (RT0 and YT9, respectively), and the Mg content was between 20.56 mg/100 g (RT0) and 174.44 mg/100 g (YT9). When comparing different types of maize, red maize presented lower values, followed by blue maize, and yellow maize, except in Ca, where blue maize *tortillas* presented higher values. When comparing different concentrations of seaweed, higher concentrations of *M. pyrifera* led to a significantly higher concentration of all minerals. 

The presence of alginate, alginic acid, and alginic acid salt in brown seaweeds such as *M. pyrifera* promotes higher mineral absorption rates than other algae because these polysaccharides have an affinity with Na, Ca, K, and Mg salts [36]. The addition of *Laminaria ochroleuca* to pasta led to an increase in minerals like Ca, P, K, and Mg [34]. However, the pasta showed lower values of Ca, P, K, and Mg (15.3, 37.9, 160.4 and 16.8 mg/100 g, respectively) than all analyzed *tortillas*. When compared to the Ca content of white and blue maize handmade *tortillas* (155 and 136 mg/100 g, respectively) [37], only the RT0 presented a slightly lower value. The addition of *Sphenarium purpuracens* [18] and nopal powder [11] also led to an increase in the mineral content. According to the Dietary Guidelines for Americans [35], the Recommended Dietary Allowance (RDA) of Ca, P, and Mg for men and women from 14 years old and more is 1000–1300 mg, 700–1250 mg, and 360–420 mg, respectively, while the Adequate Intake (AI) of K is 2300–3400 and the Chronic Disease Risk Reduction Level (CDRR) of Na is a maximum of 2300 mg. One hundred grams (roughly three *tortillas*) of BT9 would cover between 47 and 60% of the Ca RDA. One hundred grams of RT9 would cover between 26–47% and 41–48% of the RDA for P and Mg, respectively, as well as between 50 and 73% of the K AI. The Na intake would be 13% of the CDRR.

### 3.3. Quality Characteristics of Masas and Tortillas

On average, the cooked *tortillas* weighed 24.54 ± 1.02 g and had a diameter of 11.00 ± 0.79 cm and a thickness of 0.32 ± 0.09 cm. The color of the *masas* is shown in Table 4. All analyzed parameters were significantly affected by the concentration of the seaweed used. The color statistical analysis was done by comparing the different concentrations of seaweed for each type of maize, though different types of maize were not compared among them. The L* value, which represents the lightness (0-black, 100-white) in yellow and blue *masas*, significantly decreased with an increase in seaweed. In red *masa*, the incorporation of 9% seaweed was the only amount that led to a significant decrease. The a* value (redness (+) or greenness (−) in red *masa* decreased significantly with the incorporation of seaweed; similarly, YM0 presented the highest a* value. In blue *masa,* the control (BM0) presented a significantly lower value than BM3 and BM6, but a higher value than BM9. Both YM0 and RM0 presented the lowest b* values, which denotes yellowness (+) or blueness (−), while BM0, BM3, and BM6 presented no significant differences. While the hue of all control samples was significantly lower than in the samples with seaweed, chroma of YM0 and RM0 was also significantly lower and BM0 presented no significant differences with BM3 and BM6. Finally, the difference (ΔE*) between the control sample and samples with seaweed was significantly higher when seaweed concentration increased. Similarly, Alvarez-Poblano et al. [5] reported a significant decrease in L* when the concentration of *muicle* increased. However, unlike this study, they observed increased redness (higher a* values) and blueness (lower b* values).

The results of the colorimetric analysis of *tortillas* are shown in Table 5. Similar to *masas*, there was a significant decrease in L* and b* when there was an increase in seaweed concentration, which can be interpreted as the samples turning darker and increasing their blue tonalities. The incorporation of the seaweed also led to an increase in green tonalities (lower a* values). Unlike *masas*, the hue of YT0 and RT0 were significantly higher than the remaining samples, while BT3 presented a higher hue value than BT0. The chroma of all control samples (YT0, BT0, and RT0) was higher than *tortillas* with different concentrations of seaweed, which was not observed in *masas*. However, the difference (ΔE*) between the control samples (YT0, BT0, and RT0) and samples with seaweed presented the same behavior as the *masas*, in which the higher concentrations of seaweed led to a significantly higher difference. It has been reported that adding different ingredients to *tortillas*, such as soybean and amaranth, can modify the color and lead to darker *tortillas* [28], which was observed in this study. 

Additionally, when *masas* and *tortillas* were compared, L* values decreased after the cooking process and a* increased, as previously reported by Alvarez-Poblano et al. [5]. However, they also observed a decrease in b*, which was not observed for all samples in this study. It has been reported that the color of the *tortillas* depends on the type of maize and the interactions between the chemical components and the alkaline pH present during nixtamalization [38].

The texture of the *masas* (hardness) is shown in Figure 1. Hardness ranged between 2.18 and 22.32 N, with YM9 as the sample with the significantly highest value. It should be noted that the hardness of *masas* increased significantly (*p* < 0.05) with the addition of seaweed. The type of maize, as well as the interaction between the concentration of seaweed and the type of maize, was significative (*p* < 0.05). In general, *masas* prepared from yellow maize presented higher hardness. When the hardness of samples with different colors of maize but the same concentrations of seaweed was compared, yellow maize presented significantly higher values, followed by red, and subsequently blue maize. 

Contrary to the results found in this study, Argüello-García et al. [28] found that increasing the content of *J. curcas* L. flour led to a decrease in hardness, which was attributed to a higher protein content. *Masas* prepared with corn flour and *J. curcas* L. flour presented lower values of hardness (2.30–3.13 N [28], similar to nixtamalized corn *masa* with *Sphenarium purpurascens* flour (0.59–0.81 N) [18]. Nixtamalized maize *masas* obtained using several nixtamalization processes presented similar values (4–15 N) [13]. 

The results of the *tortillas*’ perforation and rollability are shown in Table 6. The perforation test allows for the measurement of the hardness of *tortillas*, which is associated with starch retrogradation and begins when *tortillas* start to cool [28]. Hardness was significantly affected (*p* < 0.05) by the type of maize and seaweed concentration, as well as the interaction between both factors. The values ranged between 1.40 N for BT3 and 4.55 N for YT9. Neither the perforation test nor the rollability showed a linear trend depending on the concentration of seaweed or based on the type of maize. However, there was a clear improvement in the rollability results when *tortillas* were heated. When *tortillas* were rolled at RT, 7 out of the 12 samples presented at least 50% breakage; after heating, only 2 samples presented a breakage of 50% and most samples presented no breakage at all. Texture in maize *tortillas* can be related to differences between grains because different maize types provide different rheological properties to the final product [32]. Texture can also be modified depending on the nixtamalization process and cooking conditions [39]. The interaction between phenolic compounds and proteins can also influence the water binding capacity and the protein–protein interactions, both of which affect the texture [32]. 

The results of several studies showed mixed results; some report an improvement in hardness, whereas others mention that the incorporation of different ingredients led to lower values. For example, the fortification of white maize *tortillas* with 10–15% chickpea hydrolysates led to higher hardness values than the control. In comparison, the fortification of blue maize *tortillas* with 15% chickpea hydrolysates led to a lower hardness than the control [32]. Argüello-García et al. [28] reported that adding 15% of *J. curcas* flour led to a hardness reduction of more than half when compared to the control, and Contreras Jiménez et al. [18] also observed higher hardness values in the control *tortillas* than in *tortillas* with *S. purpuracens*. 

With the exception of Lecuona-Villanueva et al. [40], who reported no difference between the control and *tortillas* supplemented with lysine, tryptophan, and protein concentrate from *P. lunatus*, most authors reported that the fortification of *tortillas* lead to an improvement of rollability, while the incorporation of 15% of *J. curcas* flour decreased the hardness and doubled the rollability as it increased the water retention [28]. The addition of different gums can also help increase, or at least maintain, rollability during storage [29], an effect that has also been attributed to the higher water retention ability [41].

### 3.4. Total Phenolic Content and Antioxidant Capacity of Masas and Tortillas

The TPC of *masas*, water, and methanol:water extracts are shown in Figure 2, while the results of the TPC of *tortilla* extracts in water and methanol:water are presented in Figure 3. *Masas* and *tortillas* in both water and methanol:water extracts were significantly affected (*p* < 0.05) by the type of maize, seaweed concentration, and the interaction between both factors. The TPC of *masas* in water ranged between 1141.59 and 3323.48 mg GAE/100 g *masa*, while the values in methanol:water, which were lower, were 641.89–1061.12 mg GAE/100 g *masa*. In both extracts, BM9 showed the highest phenolic content, while RM0 and YM0 presented the lowest values in methanol:water and the latter also had the lowest value in water. It can be noted that, in the water extracts, the TPC increased with the incorporation of seaweed, a trend that could not be observed in the methanol:water extracts. Compared to the results of *masas*, the obtained values for *tortillas* were lower. This contrasts with other studies that have reported that there are no significant differences among *masas* and *tortillas*. At the same time, losses of up to ~50% of the TPC are observed after lime-cooking and steeping [16]. For the *tortilla* (water extracts), the results were 838.06–2142.34 mg GAE/100 g; for the methanol:water extracts, the values ranged between 330.97 and 1018.99 mg GAE/100 g. Unlike *masas*, YT9 presented the highest and lowest values in water and methanol:water, respectively, RT0 and BT6 had the lowest TPC when extracted in water and presented no differences among them, and BT9 showed the highest value in methanol:water; the incorporation of seaweed did not show a trend in the TPC for either extract. This was in opposition to what Alvarez-Poblano et al. [5] reported, who observed an increase in TPC with higher concentrations of muicle extracts. Other studies have found contrasting results because some authors have reported that yellow maize contains lower values of TPC than other varieties of maize [9] because color can be related to the phenolic content, especially in the accumulation of anthocyanins [42], which was in accordance with the results obtained in this study. In contrast, other authors have reported no significant differences among yellow, blue, and red maize kernels [10].

The results of the antioxidant capacity (ORAC) of *masa* in both extracts are shown in Figure 4a,b, and the antioxidant capacity of *tortillas* is presented in Figure 5a,b. Similar to the TPC, *masas* and *tortillas* in both water and methanol:water extracts were significantly affected (*p* < 0.05) by the type of maize, seaweed concentration, and the interaction between both factors. The value of the *masa* water extracts were 251.40–1023.60 µmol TE/100 g *masa,* while the results of the *masa* in methanol:water extracts ranged between 14,051.96 and 44,928.75 µmol TE/100 g *masa*. Opposite to the TPC, the values of the water extracts were lower than the ones of methanol:water. In both extracts, BM9 showed the highest antioxidant capacity, while YM0 presented the lowest antioxidant capacity in water. In the methanol: water extract, YM0 and RM0 presented the lowest values and did not show significative differences. Similar to the TPC, the antioxidant capacity of water extracts increased (except in YM6 and YM9) with the incorporation of seaweed, which could also be observed for the methanol:water extracts. The antioxidant capacity, quantified as ORAC, of yellow maize has also been reported to be lower, while no differences were observed among red and blue maize [9]. Darker colored maize can also present higher antioxidant values [42] because anthocyanins are stronger antioxidants than the phenolic acids found in lighter maize [38]. However, other authors have reported no differences in the antioxidant capacity among *tortillas* made from either yellow or blue maize [37]. The results of the *tortilla* water extracts (Figure 5a) were 435.47–1738.61 µmol TE/100 g *tortillas* and the results of the *tortilla* methanol:water (Figure 5b) extracts were 14,631.47–47,327.69 µmol TE/100 g *tortillas.* In both extracts, BT9 and RT9 showed no significant differences between them. Similar to the antioxidant capacity of *masas*, the results of the methanol:water extracts were considerably higher than those of water. It could also be observed that the incorporation of seaweed increased the antioxidant capacity in almost all samples, which is in accordance to the results of Alvarez-Poblano et al. [5], who reported an increase in antioxidant activity with an increase in muicle concentration. The processing of *masa* into *tortillas* can increase the antioxidant capacity because it can help increase the soluble phenolics from bound phytochemicals [16]. The differences in the TPC and antioxidant capacities observed in both extracts for *masas* and *tortillas* could be related to the affinity of the extracted compounds. How different solvents affect the TPC has been previously investigated. For example, hot water has been proven to be more effective than organic solvents, especially if phenolic acids are the most abundant phenolic compounds [43]. However, there is not always a correlation between phenolic compounds and antioxidant capacity [43,44], which was a situation that was observed in this study. The antioxidant capacity can be influenced by the mechanism of the assay because electron transfer (ET)-based assays and hydrogen atom transfer (HAT)-based assays, like ORAC, perform differently depending on the solvent used. Polar solvents, like water, can have lower values due to hydrogen bonding, which can cause changes in the H-atom donor activities of phenolic antioxidants [45]. Another possibility could be that the methanol:water extract promoted the extraction of carotenoids, while the water extract did not, because most carotenoids are insoluble in water but soluble in organic solvents [46].

## 4. Conclusions

The incorporation of *Macrocystis pyrifera* into yellow, blue, and red maize *masas* and *tortillas* significantly affected all analyzed variables. While there were no significant changes in the carbohydrate and lipid contents, there was a decrease in moisture and an increase in ash, protein, and dietary fiber contents in relation to the incorporation of seaweed. All analyzed minerals increased. *Tortillas* prepared with red maize showed the highest concentrations, with the exception of Ca, which was higher in blue maize *tortillas*. *M. pyrifera* also affected all analyzed color parameters of both *masas* and *tortillas*, which increased the difference regarding the control samples when higher concentrations of seaweed were used. The hardness of both *masas* and *tortillas* increased with the incorporation of seaweed. The *masas* prepared with yellow maize also presented higher values than blue and red maize, which was not observed in *tortillas*. The TPC and antioxidant capacity also increased, but there was a decrease in TPC when *masas* and *tortillas* were compared. Finally, the solvent used for the TPC and antioxidant capacity assays also influenced the results because water led to a higher TPC, whereas methanol:water presented better results in the ORAC assays, which was attributed to the possible extraction of different phenolic compounds and carotenoids. The results of this study show that *M. pyrifera* can have a great potential in the development of functional foods.

## Figures and Tables

**Figure 1 foods-11-02627-f001:**
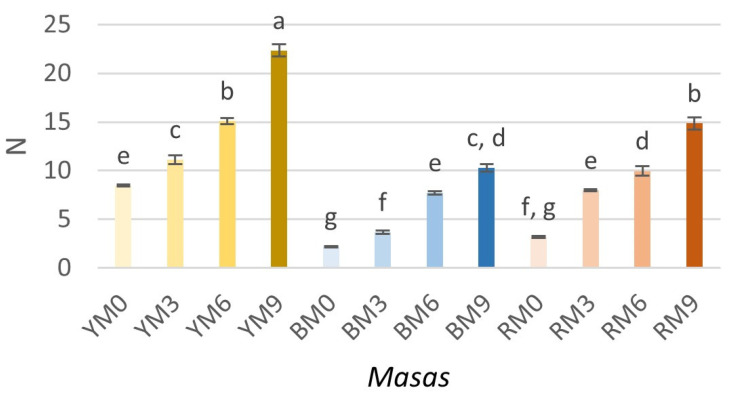
Hardness of *masas* prepared with yellow, blue, or red maize and different concentrations of brown seaweed (*Macrocystis pyrifera*). Superscripts with different letters indicate significant difference (*p* < 0.05). Results are expressed as mean (*n* = 3) and standard deviation (SD = ±). YT, BT, and RT refer to the type of maize used (yellow, blue, and red, respectively), while 0, 3, 6, and 9 denote the concentration of added seaweed (%), with 0 being the control.

**Figure 2 foods-11-02627-f002:**
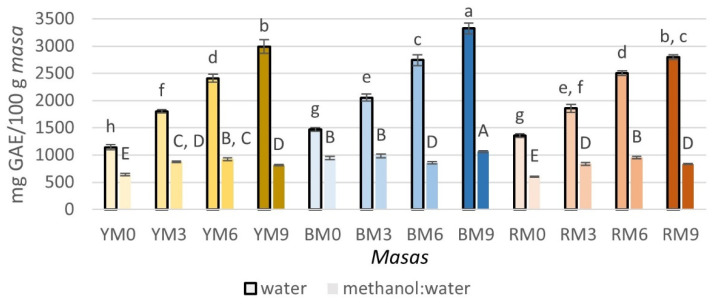
Total phenolic content (TPC) of *masas* prepared with yellow, blue, or red maize and different concentrations of brown seaweed (*Macrocystis pyrifera*). Superscripts with different letters indicate significant difference (*p* < 0.05). Results are expressed as mean (*n* = 3) ± standard deviation. YT, BT, and RT refer to the type of maize used (yellow, blue, and red, respectively), while 0, 3, 6 and 9 denote the concentration of added seaweed (%), with 0 being the control.

**Figure 3 foods-11-02627-f003:**
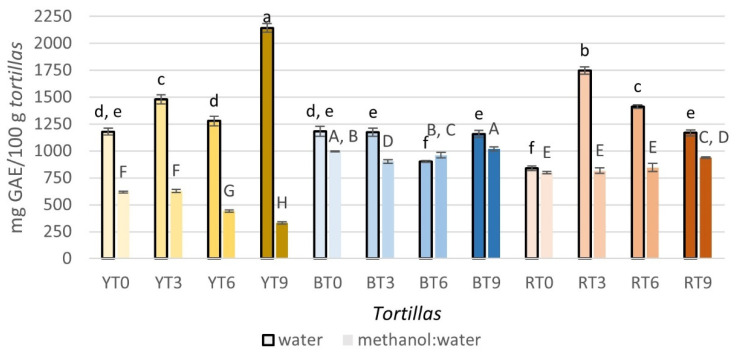
Total phenolic content (TPC) of *tortillas* prepared with yellow, blue or red maize and different concentrations of brown seaweed (*Macrocystis pyrifera*). Superscripts with different letters indicate significant difference (*p* < 0.05). Results are expressed as mean (*n* = 3) ± standard deviation. YT, BT, and RT refer to the type of maize used (yellow, blue, and red, respectively), while 0, 3, 6 and 9 denote the concentration of added seaweed (%), with 0 being the control.

**Figure 4 foods-11-02627-f004:**
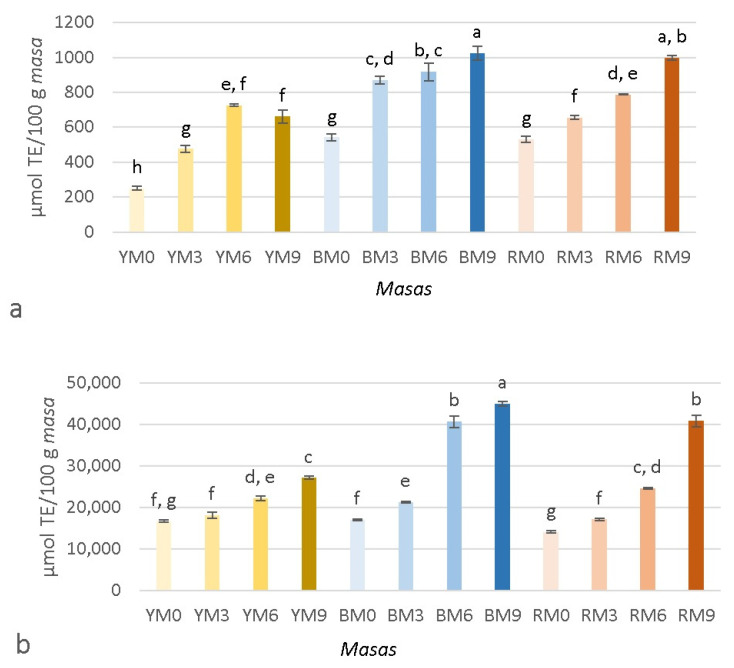
Antioxidant capacity (ORAC) of *masas* prepared with yellow, blue, or red maize and different concentrations of brown seaweed (*Macrocystis pyrifera*). (**a**) Extracts in water; (**b**) Extracts in methanol:water. Superscripts with different letters indicate significant difference (*p* < 0.05). Results are expressed as mean (*n* = 3) ± standard deviation. YT, BT, and RT refer to the type of maize used (yellow, blue, and red, respectively), while 0, 3, 6, and 9 denote the concentration of added seaweed (%), with 0 being the control.

**Figure 5 foods-11-02627-f005:**
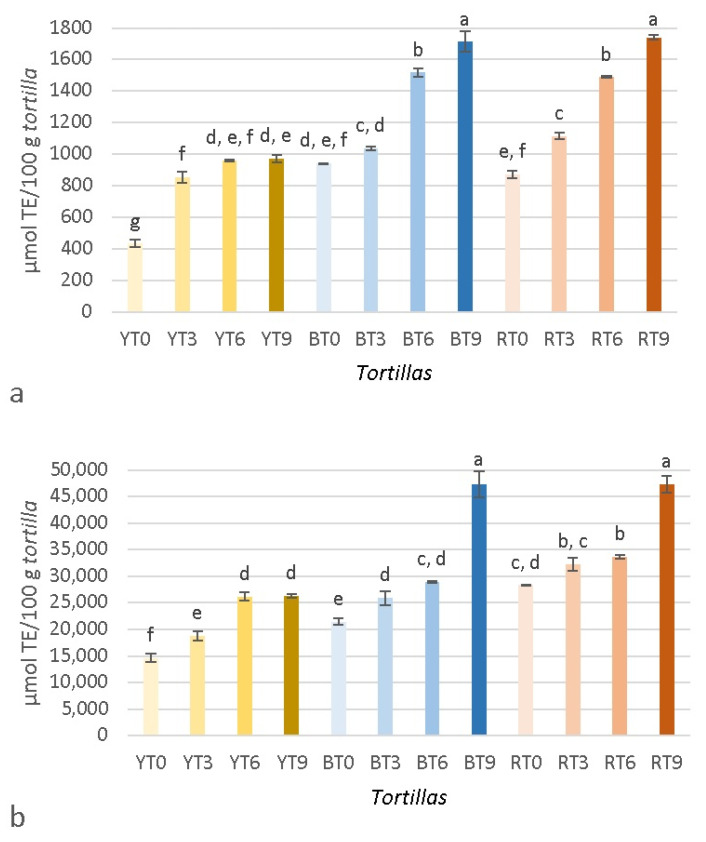
Antioxidant capacity (ORAC) of *tortillas* prepared with yellow, blue or red maize and different concentrations of brown seaweed (*Macrocystis pyrifera*). (**a**) Extracts in water; (**b**) Extracts in methanol:water. Superscripts with different letters indicate significant difference (*p* < 0.05). Results are expressed as mean (*n* = 3) ± standard deviation. YT, BT, and RT refer to the type of maize used (yellow, blue, and red, respectively), while 0, 3, 6, and 9 denote the concentration of added seaweed (%), with 0 being the control.

**Table 1 foods-11-02627-t001:** Concentration of brown seaweed (*Macrocystis pyrifera*) used and codes for each type of *masa*.

Type of Nixtamalized *Masa*						
Seaweed concentration (%)	Yellow maize		Blue maize		Red maize	
	YM0	0	BM0	0	RM0	0
	YM3	3	BM3	3	RM3	3
	YM6	6	BM6	6	RM6	6
	YM9	9	BM9	9	RM9	9

YM, BM, and RM stand for yellow, blue, and red *masa*, respectively. 0–9 are the percentages of incorporated seaweed (*w*/*w*).

**Table 2 foods-11-02627-t002:** Nutritional composition of *tortillas* with different brown seaweed (*Macrocystis pyrifera*) concentrations.

	Moisture Content (%)	Ash (%)	Protein (%)	Lipids (%)	Carbohydrates (%)	Total Fiber * (%)
YT0	44.1 ± 0.17 ^c,d^	1.16 ± 0.02 ^h^	4.49 ± 0.02 ^h^	0.36 ± 0.01 ^d^	49.9 ± 0.23 ^a^	7.01 ± 0.48 ^i,j^
YT3	43.0 ± 0.17 ^e,f^	2.20 ± 0.02 ^f^	4.56 ± 0.02 ^g,h^	0.36 ± 0.01 ^d^	49.8 ± 0.22 ^a^	7.96 ± 0.47 ^g,h^
YT6	41.9 ± 0.16 ^g^	3.24 ± 0.03 ^d^	4.64 ± 0.02 ^f,g^	0.35 ± 0.1 ^d^	49.8 ± 0.21 ^a^	8.92 ± 0.45 ^d,e,f^
YT9	40.9 ± 0.16 ^h^	4.28 ± 0.04 ^b^	4.72 ± 0.02 ^e,f^	0.35 ± 0.01 ^d^	49.8 ± 0.20 ^a^	9.88 ± 0.44 ^b,c^
BT0	46.7 ± 0.65 ^a^	1.31 ± 0.02 ^g^	4.65 ± 0.01 ^f,g^	0.55 ± 0.01 ^b,c^	46.8 ± 0.61 ^d^	6.72 ± 0.39 ^j^
BT3	45.5 ± 0.63 ^b^	2.35 ± 0.02 ^e^	4.72 ± 0.01 ^e,f^	0.54 ± 0.01 ^c^	46.9 ± 0.59 ^d^	7.68 ± 0.38 ^h,i^
BT6	44.4 ± 0.61 ^c^	3.39 ± 0.03 ^c^	4.80 ± 0.01 ^d,e^	0.53 ± 0.1 ^c^	46.9 ± 0.57 ^d^	8.65 ± 0.37 ^e,f,g^
BT9	43.2 ± 0.59 ^e^	4.42 ± 0.04 ^a^	4.87 ± 0.01 ^c,d^	0.52 ± 0.01 ^c^	46.9 ± 0.55 ^c,d^	9.62 ± 0.36 ^b,c,d^
RT0	45.5 ± 0.05 ^b^	1.12 ± 0.01 ^h^	4.88 ± 0.06 ^c,d^	0.62 ± 0.03 ^a^	47.8 ± 0.15 ^b,c^	8.24 ± 0.02 ^f,g,h^
RT3	44.4 ± 0.05 ^c^	2.16 ± 0.02 ^f^	4.95 ± 0.06 ^b,c^	0.60 ± 0.03 ^a^	47.9 ± 0.15 ^b,c^	9.16 ± 0.02 ^c,d,e^
RT6	43.3 ± 0.05 ^d, e^	3.21 ± 0.02 ^d^	5.01 ± 0.05 ^a,b^	0.59 ± 0.03 ^a^	47.9 ± 0.14 ^b^	10.1 ± 0.04 ^b^
RT9	42.1 ± 0.05 ^f,g^	4.25 ± 0.03 ^b^	5.08 ± 0.05 ^a^	0.58 ± 0.03 ^a,b^	47.9 ± 0.14 ^b^	11.0 ± 0.05 ^a^

Superscripts with different letters in the same column indicate significant difference (*p* < 0.05). Results are expressed as mean (*n* = 3) ± standard deviation. YT, BT, and RT refer to the type of maize used (yellow, blue, and red, respectively), while 0, 3, 6, and 9 denote the concentration of added seaweed (%), with 0 being the control. * Is determined as total dietary fiber.

**Table 3 foods-11-02627-t003:** Mineral content of *tortillas* prepared with yellow, blue, or red maize and different concentrations of brown seaweed (*Macrocystis pyrifera*).

	Na (mg/100 g)	Ca (mg/100 g)	P (mg/100 g)	K (mg/100 g)	Mg (mg/100 g)
YT0	10.37 ± 0.26 ^j^	362.79 ± 3.11 ^h^	314.65 ± 3.60 ^c^	331.38 ± 0.11 ^j^	84.67 ± 0.01 ^g^
YT3	107.94 ± 0.28 ^g^	403.85 ± 3.03 ^g^	319.12 ± 3.49 ^b,c^	787.79 ± 0.59 ^g^	114.60 ± 0.21 ^e^
YT6	205.50 ± 0.37 ^d^	444.92 ± 2.96 ^f^	323.60 ± 3.39 ^a,b^	1244.21 ± 1.17 ^d^	144.52 ± 0.42 ^c^
YT9	303.07 ± 0.48 ^a^	485.98 ± 2.92 ^e^	328.07 ± 3.28 ^a^	1700.62 ± 1.75 ^a^	174.44 ± 0.63 ^a^
BT0	6.85 ± 0.01 ^k^	501.57 ± 1.03 ^d^	196.76 ± 3.57 ^g^	154.35 ± 4.87 ^k^	58.38 ± 1.53 ^h^
BT3	104.52 ± 0.14 ^h^	538.46 ± 1.03 ^c^	204.77 ± 3.47 ^f^	616.08 ± 4.76 ^h^	89.09 ± 1.49 ^f^
BT6	202.19 ± 0.28 ^e^	575.36 ± 1.08 ^b^	212.78 ± 3.36 ^e^	1077.80 ± 4.72 ^e^	119.80 ± 1.49 ^d^
BT9	299.86 ± 0.41 ^b^	612.26 ± 1.17 ^a^	220.79 ± 3.25 ^d^	1539.53 ± 4.76 ^b^	150.51 ± 1.52 ^b^
RT0	3.67 ± 0.18 ^l^	100.14 ± 0.78 ^l^	66.48 ± 1.13 ^k^	53.35 ± 0.12 ^l^	20.56 ± 0.38 ^j^
RT3	101.46 ± 0.22 ^i^	149.08 ± 0.80 ^k^	78.39 ± 1.09 ^j^	518.11 ± 0.59 ^i^	52.40 ± 0.43 ^i^
RT6	199.22 ± 0.32 ^f^	198.02 ± 0.87 ^j^	90.31 ± 1.06 ^i^	982.87 ± 1.17 ^f^	84.25 ± 0.55 ^g^
RT9	296.99 ± 0.44 ^c^	246.96 ± 1.00 ^i^	102.23 ± 1.02 ^h^	1447.62 ± 1.75 ^c^	116.10 ± 0.72 ^e^

Superscripts with different letters in the same column indicate significant difference (*p* < 0.05). Results are expressed as mean (*n* = 3) ± standard deviation. YT, BT, and RT refer to the type of maize used (yellow, blue, and red, respectively), while 0, 3, 6, and 9 denote the concentration of added seaweed (%), with 0 being the control.

**Table 4 foods-11-02627-t004:** Color of *masas* prepared with yellow, blue, or red maize and different concentrations of brown seaweed (*Macrocystis pyrifera*).

	L*	a*	b*	Hue	Chroma	ΔE*	
YM0	85.72 ± 0.96 ^a^	0.14 ± 0.01 ^a^	16.91 ± 0.44 ^c^	89.51 ± 0.01 ^b^	16.91 ± 0.44 ^c^	DNA **	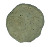
YM3	74.37 ± 0.52 ^b^	−2.44 ± 0.11 ^b,c^	21.48 ± 0.42 ^a^	96.48 ± 0.35 ^a^	21.62 ± 0.42 ^a^	12.48 ± 0.58 ^c^	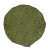
YM6	68.76 ± 0.69 ^c^	−2.65 ± 0.12 ^c^	21.65 ± 0.32 ^a^	96.98 ± 0.38 ^a^	21.81 ± 0.31 ^a^	17.80 ± 0.63 ^b^	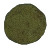
YM9	65.53 ± 0.42 ^d^	−2.35 ± 0.12 ^b^	18.71 ± 0.42 ^b^	97.15 ± 0.28 ^a^	18.86 ± 0.42 ^b^	20.40 ± 0.37 ^a^	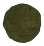
BM0	66.69 ± 0.39 ^a^	−1.48 ± 0.03 ^b^	17.35 ± 0.43 ^a^	94.88 ± 0.11 ^b^	17.42 ± 0.43 ^a^	DNA **	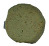
BM3	63.08 ± 0.09 ^b^	−1.1 ± 0.01 ^a^	16.52 ± 0.06 ^a,b^	93.81 ± 0.02 ^c^	16.55 ± 0.06 ^a,b^	3.44 ± 0.07 ^c^	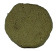
BM6	57.02 ± 0.33 ^c^	−1.15 ± 0.05 ^a^	16.98 ± 0.67 ^a,b^	93.87 ± 0.06 ^c^	17.02 ± 0.67 ^a,b^	9.33 ± 0.30 ^b^	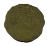
BM9	54.56 ± 0.58 ^d^	−1.79 ± 0.07 ^c^	16.06 ± 0.41 ^b^	96.38 ± 0.41 ^a^	16.17 ± 0.40 ^b^	11.85 ± 0.59 ^a^	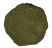
RM0	67.52 ± 1.25 ^a^	5.09 ± 0.24 ^a^	12.43 ± 0.35 ^c^	67.73 ± 1.46 ^d^	13.44 ± 0.25 ^c^	DNA **	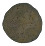
RM3	68.67 ± 1.15 ^a^	1.87 ± 0.09 ^b^	15.18 ± 0.28 ^b^	82.97 ± 0.19 ^c^	15.30 ± 0.29 ^b^	4.76 ± 0.17 ^b^	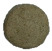
RM6	67.27 ± 0.30 ^a^	0.81 ± 0.03 ^c^	12.81 ± 0.25 ^c^	86.37 ± 0.06 ^b^	12.84 ± 0.25 ^c^	4.90 ± 0.10 ^b^	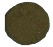
RM9	61.42 ± 0.23 ^b^	0.39 ± 0.02 ^d^	16.33 ± 0.44 ^a^	88.64 ± 0.07 ^a^	16.33 ± 0.44 ^a^	9.95 ± 0.36 ^a^	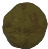

Superscripts with different letters in the same column indicate significant difference (*p* < 0.05). Results are expressed as mean (*n* = 3) ± standard deviation. YM, BM, and RM refers to the type of maize used (yellow, blue, and red, respectively), while 0, 3, 6, and 9 denote the concentration of added seaweed (%), with 0 being the control. DNA ** does not apply. The statistical analysis was performed only between samples of the same type of maize.

**Table 5 foods-11-02627-t005:** Color of *tortillas* prepared with yellow, blue, or red maize and different concentrations of brown seaweed (*Macrocystis pyrifera*).

	L*	a*	b*	Hue	Chroma	ΔE*	
YT0	66.28 ± 0.86 ^a^	0.79 ± 0.03 ^d^	23.33 ± 0.39 ^a^	88.05 ± 0.05 ^a^	23.34 ± 0.39 ^a^	DNA **	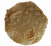
YT3	44.40 ± 0.68 ^c^	4.38 ± 0.12 ^a^	19.99 ± 0.71 ^b^	77.63 ± 0.74 ^d^	20.46 ± 0.67 ^b^	21.53 ± 0.75 ^c^	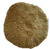
YT6	49.98 ± 0.41 ^b^	1.44 ± 0.05 ^c^	21.25 ± 0.73 ^b^	86.12 ± 0.19 ^b^	21.30 ± 0.73 ^b^	15.54 ± 0.50 ^b^	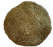
YT9	40.90 ± 1.29 ^d^	3.42 ± 0.05 ^b^	17.17 ± 0.45 ^c^	78.74 ± 0.20 ^c^	17.51 ± 0.45 ^c^	25.36 ± 1.26 ^a^	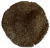
BT0	49.40 ± 1.19 ^a^	6.70 ± 0.19 ^a^	18.55 ± 0.41 ^a^	70.13 ± 0.44 ^b^	19.72 ± 0.43 ^a^	DNA**	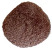
BT3	44.34 ± 0.19 ^b^	4.30 ± 0.04 ^b^	15.71 ± 0.21 ^b^	74.69 ± 0.23 ^a^	16.29 ± 0.21 ^b^	5.77 ± 0.14 ^c^	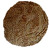
BT6	40.88 ± 0.27 ^c^	3.75 ± 0.05 ^c^	12.71 ± 0.30 ^c^	73.55 ± 0.44 ^a^	13.25 ± 0.29 ^c^	10.22 ± 0.38 ^b^	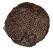
BT9	35.82 ± 0.73 ^d^	3.03 ± 0.14 ^d^	6.42 ± 0.17 ^d^	64.70 ± 1.34 ^c^	7.10 ± 0.14 ^d^	18.05 ± 0.59 ^a^	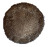
RT0	52.39 ± 1.73 ^a^	6.35 ± 0.31 ^a^	16.53 ± 0.26 ^a^	68.98 ± 1.12 ^a^	17.71 ± 0.20 ^a^	DNA**	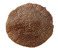
RT3	44.48 ± 0.14 ^b^	5.07 ± 0.11 ^c^	14.76 ± 0.09 ^b^	71.03 ± 0.42 ^a^	15.61 ± 0.08 ^b^	7.28 ± 0.14 ^c^	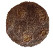
RT6	40.04 ± 0.44 ^c^	5.47 ± 0.19 ^b,c^	12.53 ± 0.29 ^c^	66.41 ± 0.61 ^b^	13.67 ± 0.31 ^c^	12.13 ± 0.32 ^b^	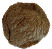
RT9	37.30 ± 0.29 ^d^	6.00 ± 0.18 ^a,b^	10.52 ± 0.26 ^d^	60.29 ± 1.33 ^c^	12.12 ± 0.14 ^d^	15.49 ± 0.35 ^a^	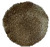

Superscripts with different letters in the same column indicate significant difference (*p* < 0.05). Results are expressed as mean (*n* = 3) ± standard deviation. YT, BT, and RT refer to the type of maize used (yellow, blue and red, respectively), while 0, 3, 6, and 9 denote the concentration of added seaweed (%), with 0 being the control. DNA ** does not apply. The statistical analysis was performed only between samples of the same type of maize.

**Table 6 foods-11-02627-t006:** Texture analysis of *tortillas* prepared with yellow, blue, or red maize and different concentrations of brown seaweed (*Macrocystis pyrifera*).

	Perforation (N)	Rollability *	
		Room Temperature (RT) *Tortillas*	Heated *Tortillas*
YT0	2.61 ± 0.11 ^e,f^	5 ± 0.00 ^a^	3 ± 0.00 ^a^
YT3	3.43 ± 0.13 ^b^	4 ± 0.00 ^b^	2 ± 0.00 ^b^
YT6	2.30 ± 0.08 ^g^	2 ± 0.00 ^d^	1 ± 0.00 ^c^
YT9	4.55 ± 0.09 ^a^	5 ± 0.00 ^a^	2 ± 0.00 ^b^
BT0	3.30 ± 0.12 ^b,c^	1 ± 0.00 ^e^	1 ± 0.00 ^c^
BT3	1.40 ± 0.05 ^h^	3 ± 0.00 ^d^	1 ± 0.00 ^c^
BT6	1.62 ± 0.04 ^h^	5 ± 0.00 ^a^	1 ± 0.00 ^c^
BT9	1.49 ± 0.03 ^h^	5 ± 0.00 ^b^	1 ± 0.00 ^a^
RT0	3.10 ± 0.11 ^c,d^	1 ± 0.00 ^e^	1 ± 0.00 ^c^
RT3	2.50 ± 0.09 ^f,g^	2 ± 0.00 ^c^	1 ± 0.00 ^c^
RT6	2.27 ± 0.09 ^g^	5 ± 0.00 ^a^	1 ± 0.00 ^c^
RT9	2.83 ± 0.03 ^d, e^	4 ± 0.00 ^a^	3 ± 0.00 ^c^

Superscripts with different letters in the same column indicate significant difference (*p* < 0.05). Results are expressed as mean (*n* = 3) ± standard deviation. YT, BT, and RT refer to the type of maize used (yellow, blue, and red, respectively), while 0, 3, 6 and 9 denote the concentration of added seaweed (%), with 0 being the control. * Subjective scale where 1 = 0%, 2 = 25%, 3 = 50%, 4 = 75%, and 5 = 100% breakage.

## Data Availability

The data presented in this study are available in this article.

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
