# Peer review of "Effect of Brown Seaweed (Macrocystis pyrifera) Addition on Nutritional and Quality Characteristics of Yellow, Blue, and Red Maize Tortillas"

_foods, 2022, doi:10.3390/foods11172627_

Round 1

Reviewer 1 Report

Comments and Suggestions for Authors

Review on manuscript: foods-1817666

Effect of brown seaweed (Macrocystis pyrifera) addition on nutritional and quality characteristics of yellow, red, and blue maize

by Alexa Pérez-Alva, Diana K. Baigts-Allende, Melissa A. Ramírez-Rodrigues and Milena M. Ramírez-Rodrigues

submitted to Foods

In the manuscript submitted for comments, the authors studied the effect of brown seaweed (Macrocystis pyrifera) addition on nutritional and quality characteristics of maize-based products.

In my opinion, the manuscript requires many additions and explanations, the purpose of the work is unclear, and the description of preparation and processing of the tested products and methods used is incomplete. Therefore, the manuscript requires major revision.

Detailed recommendation:

Title – the title of the manuscript is unclear, the authors analyzed the effect of brown seaweed on maize-based products and not on the maize itself,

lines 14-15 – the results presented in this way do not show the effect of brown seaweed additive,

line 29 – should be: mg GAE/100 g,

lines 69-72 – the purpose of the work, as well as its title, is unclear, reference to the methods used is not necessary here,

lines 74-77 – maize mass and seaweed are not chemicals or standards, it is a test material,

lines 81-90 – in this paragraph the authors should clearly describe how the masses and tortillas have been prepared and thermally processed and the difference between them,

line 86 – heating conditions should be specified,

line 88 – what exactly does room temperature mean?

line 90 – freezing conditions should be specified,

line 92 – Latin names should be italicized,

Table 1 – is it really needed? this information may be provided in the text,

line 96 – the protein conversion factor should be given,

line 96 – crude or dietary fiber?

line 112 – origin country should be specified,

line 115 – the type of illuminant and the measurement geometry should be given,

line 127 – TPA test consists in double compressing of the tested material, did the authors really devise such a procedure? how was the tortilla compressed?

line 130 – what texture parameters were determined? what are their definitions?

line 134 – what device was used to monitor fluorescence?

lines 155-160 – authors should consider using two-way ANOVA - maize type and additive level,

line 192 – linguistic proofreading is needed,

Table 2 – crude or dietary fiber?

lines 207-233 – authors should limit the repetition of numerical values from the table,

Tables 4 and 5 – are chroma values needed here? they are very close to b *,

Figure 1 – Y axis description is missing,

lines 309-312 – Newton is not a unit of adhesiveness,

lines 311-312, 319, 337, 381, 387 and next – Latin names should be italicized,

line 318 – should be: Figure 2 - Y axis description is missing, the values of adhesiveness given by the authors raise doubts, if the adhesiveness is related to the area of the negative peak on the curve, why are some of the authors' results positive and others negative?

lines 357-358 – should be: mg GAE,

Figures 3-6 – Y axis description is missing,

lines 515, 523, 547, 574 – incorrect name of the journals.

Reviewer 2 Report

Comments and Suggestions for Authors

The article deals with a subject of interest for the food industry and the development of new functional products. However, the following issues should be clarified:

- the introduction should contain some additional data about the nutritional value and bioactive compounds of the algae.

- line 86: please add the product processing parameters, such as the baking temperature.

- How did you choose the addition percentages of 3, 6 and 9?

- Considering that you have created a product that is addressed to consumers, it is necessary to add a sensory analysis. Analyzing the degree of acceptability is important because in this way you could decide what is the optimal percentage of algae and if the recipes need to be modified.

- Have you analyzed the chemical composition of the algae used? How do you explain such a large increase in nutrients and antioxidant activity when adding such small percentages of algae?

Round 2

Reviewer 1 Report

Comments and Suggestions for Authors

The corrections introduced by the authors to the manuscript have significantly improved its quality, therefore the article could be accepted for publication.

Reviewer 2 Report

Comments and Suggestions for Authors

Thank you for sending me the revised form. It seems that the suggestions have been mostly resolved, therefore the article has been improved.